Measuring short distance dispersal of Alliaria petiolata and determining potential long distance dispersal mechanisms

Loebach Christopher A. chris_loebach@yahoo.com
Anderson Roger C.
School of Biological Sciences, Illinois State University , Normal , IL , United States of America
Woods Kerry
Electronic publication date: 2018 Mar 15
Publication date: 2018
Volume: 6
Electronic Location ID: e4477
Received 2017 Jul 6; Accepted 2018 Feb 19
Copyright: ©2018 Loebach and Anderson
Copyright year: 2018
Copyright holder: Loebach and Anderson
License: This is an open access article distributed under the terms of the Creative Commons Attribution License, which permits unrestricted use, distribution, reproduction and adaptation in any medium and for any purpose provided that it is properly attributed. For attribution, the original author(s), title, publication source (PeerJ) and either DOI or URL of the article must be cited.
License URL: https://creativecommons.org/licenses/by/4.0/

Keywords: Alliaria petiolata, Garlic mustard, Seed dispersal, Epizoochory, Dispersal kernal, Seed attachment, Seed retention

Funding: Phi Sigma Biological Society Robert D. Weigel Fund E.L. Mockford and C.F. Thompson Summer Research Fellowship This work was supported by the Beta Lambda Chapter of the Phi Sigma Biological Society Robert D. Weigel Fund and the E.L. Mockford and C.F. Thompson Summer Research Fellowship. The funders had no role in study design, data collection and analysis, decision to publish, or preparation of the manuscript.

==============================
Introduction

Alliaria petiolata, an herbaceous plant, has invaded woodlands in North America. Its ecology has been thoroughly studied, but an overlooked aspect of its biology is seed dispersal distances and mechanisms. We measured seed dispersal distances in the field and tested if epizoochory is a potential mechanism for long-distance seed dispersal.

Methods

Dispersal distances were measured by placing seed traps in a sector design around three seed point sources, which consisted of 15 second-year plants transplanted within a 0.25 m radius circle. Traps were placed at intervals ranging from 0.25–3.25 m from the point source. Traps remained in the field until a majority of seeds were dispersed. Eight probability density functions were fitted to seed trap counts via maximum likelihood. Epizoochory was tested as a potential seed dispersal mechanism for A. petiolata through a combination of field and laboratory experiments. To test if small mammals transport A. petiolata seeds in their fur, experimental blocks were placed around dense A. petiolata patches. Each block contained a mammal inclusion treatment (MIT) and control. The MIT consisted of a wood-frame (31 × 61× 31 cm) covered in wire mesh, except for the two 31 × 31 cm ends, placed over a germination tray filled with potting soil. A pan filled with bait was placed in the center of the tray. The control frame (11 × 31 × 61 cm) was placed over a germination tray and completely covered in wire mesh to exclude animal activity. Treatments were in the field for peak seed dispersal. In March, trays were moved to a greenhouse and A. petiolata seedlings were counted and then compared between treatments. To determine if A. petiolata seeds attach to raccoon (Procyon lotor) and white-tailed deer (Odocoileus virginianus) fur, wet and dry seeds were dropped onto wet and dry fur. Furs were rotated 180 degrees and the seeds that remained attached were counted. To measure seed retention, seeds were dropped on furs and rotated as before, then the furs were agitated for one hour. The seeds retained in the fur were counted.

Results

For the seed dispersal experiment, the 2Dt function provided the best fit and was the most biologically meaningful. It predicted that seed density rapidly declined with distance from the point source. Mean dispersal distance was 0.52 m and 95% of seeds dispersed within 1.14 m. The epizoochory field experiment showed increased mammal activity and A. petiolata seedlings in germination trays of the MIT compared to control. Laboratory studies showed 3–26% of seeds were attached and retained by raccoon and deer fur. Retention significantly increased if either seed or fur were wet (57–98%).

Discussion

Without animal seed vectors, most seeds fall within a short distance of the seed source; however, long distance dispersal may be accomplished by epizoochory. Our data are consistent with A. petiolata’s widespread distribution and development of dense clusters of the species in invaded areas.

Introduction

Alliaria petiolata, garlic mustard (Brassicaceae: Bieb. [Cavara and Grande]), is an herbaceous invasive species that has invaded woodlands in eastern North America (Anderson, Dhillion & Kelley, 1996). Alliaria petiolata is native to Eurasia, occurring from England to Sweden to Turkestan, northwestern-Himalayas, India and Sri Lanka, and south to Italy and the Mediterranean basin (Tutin et al., 1964; Cavers, Muriel & Robert, 1979). It also occurs outside of its native range in Australia (CAB International, 2015; EDDMapS, 2015). The species was first recorded in North America on Long Island, New York in 1868 where it was introduced by humans as a food plant (Nuzzo, 1993; Roberts & Anderson, 2001). Since that time, it has spread exponentially and currently occurs in 37 states that stretch from the New England area to the west coast and five Canadian provinces (USDA NRCS, 2014). It is classified as invasive in 20 US states and in the five Canadian provinces (CAB International, 2015; EDDMapS, 2015).

Alliaria petiolata has been extensively studied in an effort to understand its invasive ability and impact on native communities (Rodgers, Stinson & Finzi, 2008). To better understand A. petiolata invasive ability, studies have investigated the competition between A. petiolata and native plant species (Bauer, Anderson & Anderson, 2010; Philips-Mao, Larson & Jordan, 2014), the role of disturbance caused by Lumbricus terrestris and L. rubellus (Nuzzo, Marerz & Blossey, 2009), and the preferential browsing of white-tailed deer, Odocoileus virginianus on native species (Knight et al., 2009; Kalisz, Spigler & Horvitz, 2014). Alliaria petiolata continues to invade new areas (Welk, Schubert & Hoffmann, 2002) and persists in areas where it has become established, although its abundance in invaded areas can decline over time (Davis et al., 2012; Lankau et al., 2009). A largely overlooked aspect of A. petiolata’s biology is seed dispersal distances and mechanisms (Barney & Whitlow, 2008). Closing this knowledge gap is important for improving our understanding of the invasive ability of A. petiolata and how it disperses across the landscape.

Eschtruth & Battles (2009), Eschtruth & Battles (2011) and Eschtruth & Battles (2014) studied A. petiolata’s ability to invade new areas and found that propagule pressure is the most important factor. The importance of propagule pressure was tested through a propagule pressure model. However, this model was was built on untested dispersal distances assumptions and predicted that 95% of seeds fall within the maximum reported distance of dispersal of 2 m as described in Nuzzo (1999) and Drayton & Primack (1999). The reported dispersal distances were based on observations and simple field tests (Nuzzo, 1999; Drayton & Primack, 1999; V Nuzzo, pers. comm., 2014), not experimental data. Therefore, it is possible that the importance of propagule pressure was not accurately estimated due to the parameters of the underlying model being based on untested assumptions. Experimentally measuring dispersal distances in the field may provide the basis for a more accurate estimate of propagule pressure and its importance in A. petiolata invasion.

If the vast majority of A. petiolata seeds are dispersed within 2 m of the parent plant as reported in Nuzzo (1999) and Drayton & Primack (1999), then A. petiolata populations are predicted to spread at a rate of less than 1 m per year, which is below the observed average spread rate of 5.4 m per year (Nuzzo, 1999). In addition, A. petiolata spreads through the establishment of satellite populations that are well ahead of the invasion front (Nuzzo, 1993; Nuzzo, 1999; Burls & McClaugherty, 2008). Both the rapidly moving invasion front and the establishment of satellite populations suggest the presence of a long-distance dispersal mechanism (Nuzzo, 1993; Nuzzo, 1999; Burls & McClaugherty, 2008; Eschtruth & Battles, 2011).

Cavers, Muriel & Robert (1979) briefly discussed long-distance dispersal mechanisms of A. petiolata and stated that seeds did not float well but readily adhered to a damp cloth. Therefore, epizoochory has been suggested as a possible dispersal mechanism (Blossey et al., 2001; Cavers, Muriel & Robert, 1979; Evans et al., 2012) with deer, mice, and other small mammals possibly transporting the seed. But this hypothesis has not been explicitly tested.

The Dispersal Diaspore Database (DDD) (Hintze et al., 2013) contains seed dispersal information for over 2,111 plant species to predict and rank the epizoochory potential of these species by combining two metrics, the ability of a seed to attach to fur (Will, Maussner & Tackenberg, 2007), and to be retained in the fur once attached (Römermann, Tackenberg & Poschlod, 2005; Tackenberg et al., 2006). Of the 2,111 species in the index, 64% were better adapted to epizoochory than A. petiolata. Alliaria petiolata seeds lack any clear adaptions for epizoochory such as hooks or barbs, but they have several favorable traits including small size and partial exposure in the fruit (Hintze et al., 2013). While these results are not highly suggestive of epizoochory, they may not have captured A. petiolata’s true potential for epizoochory. Many plant species are dispersed long distances by a mechanism for which they have no apparent adaptations (Clark et al., 1998; Higgins, Nathan & Cain, 2003; Myers & Gardescu, 2004).

Studies that comprise the DDD found that attachment potential and retention potential differed among the European mammal species tested (Tackenberg et al., 2006; Will, Maussner & Tackenberg, 2007). Since epizoochory potential differs among mammal species, it is important to conduct epizoochory tests on mammal species that A. petiolata is likely to encounter in North America. The mammals mentioned in Blossey et al. (2001) and Evans et al. (2012) are logical animals to test since it was hypothesized they were vectors involved in A. petiolata long distance dispersal. Additionally, the dampness of the fur may also affect epizoochory potential. Tackenberg et al. (2006) found that fur dampness did not have a consistent effect on the retention potential for all 19 species they tested, but dampness increased retention potential for a few species. Cavers, Muriel & Robert (1979) noted that A. petiolata seeds readily adhered to a damp cloth suggesting that the seeds may be more likely to stick to damp rather than dry fur.

Our study had two objectives. The first objective was to experimentally measure A. petiolata seed dispersal distances in the field using seed traps and to use these data to estimate the parameters of eight dispersal kernels. The results of our dispersal study were then compared the dispersal function used by Eschtruth & Battles (2009), Eschtruth & Battles (2011) and Eschtruth & Battles (2014). We also used the results of our study to calculate median dispersal distance and distance at which 95% are dispersed within.

Our second objective was to test the hypothesis that epizoochory via North American woodland mammals is a long-distance seed dispersal mechanism in A. petiolata. We tested this hypothesis through field and complimentary laboratory studies. The field study was designed to attract small mammals to experimental areas to determine if high mammal activity caused these areas to accumulate more seeds resulting in higher densities of first-year A. petiolata seedlings than in areas with low mammal activity. Laboratory studies measured attachment potential and retention potential of wet and dry A. petiolata seeds applied to wet and dry fur of raccoon (Procyon lotor) and white-tailed deer pelts. Our study is the first to experimentally measure A. petiolata seed dispersal distances in the field and also to demonstrate that epizoochory is a probable long-distance seed dispersal mechanism.

Methods

Study species

Alliaria petiolata is a member of the mustard family (Brassicaceae) and is a winter biennial. Germination occurs in late winter or early spring and basal rosettes are formed the first year. During early spring of the second year, plants bolt and rapidly increase shoot length with stem elongation of up to 1.9 cm per day between the 18th of April and the 13th of May (Anderson, Dhillion & Kelley, 1996). Flowers form in March and April, while fruits develop in May and June. Seeds are small ((mean ± SE, L × W, 3.6 ± 0.05 × 1.3 ± 0.03 mm), range L (3.1–4.5 mm) and W (0.9–1.9 mm), N = 50) (Loebach & Anderson, 2017) and Mullarkey, Byers & Anderson (2013) reported that seed mass varied from 2.11 ± 0.04 to 2.38 ± 0.034) depending upon cross type (e.g., within populations, between populations, or selfing). According to Anderson, Dhillion & Kelley (1996), seeds are dispersed from July to October with peaks occurring in August and September. Baskin & Baskin (1992) found that 70% of seeds germinated in the first year under favorable conditions, but seeds can persist in the seed bank up to five years (Baskin & Baskin, 1992).

Study sites

Study sites were located within two properties of the Parklands Foundation, the Merwin Nature Preserve and South Breens Woods. The Merwin Nature Preserve is 25 km and South Breens Woods is 20 km north of Normal, IL USA. The Merwin Nature Preserve is a 325 ha oak-hickory dominated second-growth forest that has been protected from livestock grazing since the 1970’s. The South Breens Woods is a 4 ha oak dominated forest and has been under protection since 1979.

Dominant tree species at the Merwin study area are Ohio buckeye (Aesculus glabra), hackberry (Celtis occidentalisi), American elm (Ulmus americana), yellowbud hickory (Carya cordiformis), and black walnut (Juglans nigra). The dominant ground layer species are wood nettle (Laportea canadensis), black snake root (Sanicula odorata), wing stem (Verbesina alternifolia), Virginia wildrye (Elymus virginicus), and Virginia creeper (Parthenocissus quinquefolia). The mapped soil type is straw loam (224C2).

At Breens Woods, the dominant tree species are white oak (Quercus alba), American elm, red elm (Ulmus rubra), black cherry (Prunus serotina), and iron wood (Ostrya virginiana). The dominant ground species are Virginia creeper, Solomon seal (Polygonatum commutatum), false Solomon seal (Smilicina racemosa), jack in the pulpit (Arisaema triphyllum), and sweet-scented bedstraw (Galium triflorum). The mapped soil type is Birbeck silt loam (233B2) (Soil Survey of McLean County, Illinois, 2004).

Alliaria petiolata was present and abundant at both sites as were other invasive species such as buckthorn (Rhamnus cathartica) and honeysuckle (Lonicera maackii).

Seed trap design

Our experiment was designed to determine the A. petiolata dispersed seed density at increasing distances away from the seed source and to use these data to estimate the parameters of the eight dispersal kernels. A dispersal kernel is a probability density function (pdf) that describes the dispersal of seeds from a parent plant (Clark et al., 1999). There are two types of dispersal kernels, the dispersal location kernel, g(r), and the dispersal distance kernel, f(r) (Nathan et al., 2012). The g(r) describes the probability of a seed dispersing into an infinitely small area at a given distance from the parent plant and it can be used to predict the number of seeds that are expected to land in a specific area at a specific distance from the seed source (Schurr, Steinitz & Nathan, 2008). The eight g(r) described in Cousens, Dytham & Law (2008) was used in this study (Table 1). We compared the predictions of the g(r) from our study to the negative exponential function used by Eschtruth & Battles (2009), Eschtruth & Battles (2011) and Eschtruth & Battles (2014).

Table 1 The eight g(r) dispersal functions as described in Nathan et al. (2012) that were fitted to the seed trap data.

The parameter a is a shape parameter and b is a scale parameter which determines the relative weight of long distance dispersal events, and r is the distance from the center of the point source.

Function	g(r)	
Negative exponential	12πa2 exp−ra	
Log normal	12π3∕2br2 exp−logra22b2	
2Dt	b−1πa21+r2a2−b	
Weibull	b2πa2rb−2 expr2ab	
Gaussian	1πa2 expr2a2	
Logistic	b2πa2Γ2∕bΓ1−2∕b exp1+rbab−1	
Exponential power	b2πa2Γ2∕b exp−rbab	
(Inverse) power-law	b−2b−12πa21+ra−b	

The f(r) describes the probability of a seed dispersing a specific distance and was used to calculate median dispersal distance and distance at which 95% are dispersed within (Cousens, Dytham & Law, 2008).

Typically, a pdf is generated through the use of seed traps placed in a specific design around a seed source (Bullock, Shea & Skarpaas, 2006). A mathematical function describing a g(r) is fitted to the trap data to estimate the shape of the dispersal kernel. Assuming dispersal is isotropic, the same in all directions, the calculated g(r) can be converted to the f(r) with the equation: (1) fr=2πrgr.

(Cousens & Rawlinson, 2001).

The seed dispersal study was conducted at Merwin Nature Preserve. Alliaria petiolata seed point sources were established in areas where there were no trees or shrubs within 3.5 m, where understory vegetation cover was less than 20%, and where there was nearly level topography. Sites were also located within the interior of the woodlands with a full canopy. Sites were selected for these characteristics to minimize variation in dispersal distances due to the surrounding vegetation and gravity.

An A. petiolata seed point source consisted of 15 second-year A. petiolata plants transplanted into a single 0.25 m radius circle. The 15 plants were randomly located within the circle. In total, three point sources were created. Plants were transplanted during the late stages of fruit development just prior to the beginning of dehiscence. Since isolation is important for increasing the effectiveness of this experimental design (Bullock, Shea & Skarpaas, 2006), all second-year A. petiolata plants within 9 m of the point source were removed. In the area beyond the 9 m, scattered A. petiolata plants occurred, but there were no dense stands. Dispersal was assumed to be isotropic (the same in all directions). To capture the seed rain, seed traps were placed at intervals of increasing distance around the point source in a sector design, which is the most effective design for assessing isotropic dispersal (Skarpaas, Shea & Bullock, 2005). One sector consisted of 45 azimuth degrees beginning at zero degrees north for a total of eight sectors. Within a sector, traps were placed at distances 0.25, 0.50, 0.75, 1.25, 2.25, 3.25 m from the center of the point source. In each sector, one trap was placed at distances 0.25, 0.50, and 0.75 m, two traps at 1.25 m, four at 2.25 m, and six at 3.25 m from the point source (Fig. 1). The number of traps increased as a step function as distance from the point source increased to maintain a reasonable probability of capturing a seed as distance increased. The number of traps was not increased until after 0.75 m to keep the total quantity of traps to a feasible number (Bullock & Clarke, 2000).

Figure 1 A diagram of the seed trap study experiment design.

Seed traps were place at distances 0.25, 0.5, 0.75, 1.25, 2.25, and 3.25 m from the seed point source and at every 45 azimuth degrees.

Seed traps consisted of two plastic cups with diameter of 9.5 cm and height of 12 cm. One cup was placed inside the other and nylon cloth was placed between the cups. Several small holes were inserted into the bottom of both cups for water drainage while the cloth captured the seeds. Each trap was placed in a hole slightly larger than the cup’s diameter and deep enough so the top of the trap was flush with the ground surface. At distances with more than one trap, traps were placed so each touched its neighbor and all were equidistant from the center of the point source. For each point source, there was a total of 120 traps for 0.855 m2 of trapping area.

Seed traps were placed around one point source in summer 2013 (Point Source 1) and two point sources in 2014 (Point Sources 2 and 3). Traps were placed around the point source before the siliques began dehiscence and were collected after the vast majority of seeds dispersed. Traps were in the field from July 24th to October 5th and July 12th to August 28th for 2013 and 2014, respectively. After the seed traps were collected, the numbers of seeds in each trap were counted in the laboratory at Illinois State University.

The total number of seeds dispersed from a point source was estimated by subtracting the number of seeds that were not dispersed from each point source at the end of the experiment, from the estimated total at the beginning. To estimate the initial total number of seeds in a point source, the length of each silique was measured and the number of seeds inside was estimated with the equation S =  − 6.8 + 4.38x (F1,138 = 419.5, p < 0.0001, R2 = 0.752). S is seed number and x is silique length in cm (Loebach & Anderson, 2018). When seed traps were collected, the siliques remaining in the point source were also collected and the seeds within them were counted in the laboratory.

Estimating dispersal kernels

Seed count data from the three A. petiolata seed point sources were used to estimate the parameters of eight different g(r) dispersal functions that are described in Nathan et al. (2012). These functions include a scale parameter (a) and a shape parameter (b), except for the Gaussian and negative exponential functions, which only have the a parameter. Since dispersal was assumed to be isotropic, direction was ignored when fitting the g(r) functions. While there was variation in seed counts among the directions, there was no consistent pattern. Also, assuming isotropic dispersal allows for more general predictions about dispersal distances to be made than if directions were analyzed separately. Lastly, there are no known a priori reasons for why directions would differ.

For each point source, the g(r) functions were fitted to the seed count data using the following equation: (2) n=grAQ

where the parameter n was the seed number captured by a trap, g(r) was one of the eight functions evaluated at distance r, A was the area of a seed trap (0.007125 m2), and Q was the estimated number of seeds within the point source around which the trap was located. Parameter values for the dispersal functions were estimated by non-linear mixed effects modelling, which minimizes the negative log-likelihood value (−lnL) using maximum likelihood (PROC NLMIXED) in SAS® software 9.3 (SAS Institute, 2012). The default quasi-Newton algorithm was used. The product AQ was included as an offset variable as suggested by Cousens, Dytham & Law (2008). An additional random effect parameter (u) was included to account for random variation among the three point sources.

Dispersal functions were fit to the data using a log-link function and a negative binomial error distribution. A Poisson distribution was also utilized, but in all cases the negative binomial had a better fit. The negative binomial distribution assumes seeds are distributed with a mean of N and the dispersion parameter k, which accounts for over dispersion (Clark et al., 2005). The dispersal function with the lowest Akaike Information Criterion (AIC) score was selected for all further analysis (Johnson & Omland, 2004).

The selected g(r) was evaluated to ensure that it met the requirements of a pdf. These requirements are that the function must be positive over the entire expressed space and the function must integrate to one (Cousens & Rawlinson, 2001). The selected g(r) was then converted to the f(r) with (1). The f(r) was evaluated to determine if it met the requirements of a pdf (Peart, 1985). The g(r) and resultant f(r) with the lowest AIC score, and that met the requirements of a pdf, were selected.

The selected g(r) was analyzed to determine how quickly the probability of a seed being dispersed into an infinitely small area decreased as distance from the point source increased. The g(r) was used to predict the number seeds that would arrive in an area through (2). These predictions were then compared to the actual seed counts from the field. The selected g(r) function was also compared to the negative exponential from Eschtruth & Battles (2009) by using both functions to predict the change in dispersed seed density as distance increased from a single second-year A. petiolata plant. The fecundity of A. petiolata plants was set to 156 seeds as this was the fecundity value used in Eschtruth & Battles (2009). The f(r) was analyzed to calculate the median dispersal distance and the distance at which 95% of seed are dispersed by determining the distance at which the f(r) integrated to 0.50 and 0.95, respectively.

Epizoochory field experiment

To determine if epizoochory occurs in the field, we placed experimental blocks around dense patches of second-year A. petiolata plants. In the summer of 2013, blocks were established around the perimeter of three A. petiolata patches at the Merwin Nature Preserve. In 2014, blocks were established around one A. petiolata patch at Merwin Nature Preserve and at three patches at South Breens Woods. At each A. petiolata patch, one block was placed at the outer edge of the patch in each of the four cardinal directions from the patch center for a total of four blocks per patch. In total, there were 28 blocks placed around seven A. petiolata patches.

Each block contained a mammal inclusion treatment (MIT) and a control. In both treatments, a germination tray filled with potting soil was placed into the ground so it was flush with the ground surface. The MIT was designed to increase mammal activity over germination trays relative to the control. A control replicate consisted of a wood-frame (11 × 61 × 31 cm) completely covered with 1.2 cm2 size wire mesh placed over a germination tray. A MIT replicate consisted of a wooden frame (31 × 61 × 31 cm) covered with 2.5 cm mesh poultry fencing placed over a tray. The two 31 × 31 cm ends of the MIT were not covered to allow raccoon-sized or smaller animals to enter. Each frame included a shallow metal pie pan (23 cm diameter) attached to bottom in the center. Only pans in the MIT were filled daily with bait (200 ml equal parts of cracked corn and black oil sunflower seeds) to attract mammals. Within a block, the position of the MIT and control were randomly assigned and were placed 1 m apart. All second-year A. petiolata plants located within 1.5 m of the block were removed to prevent significant amounts of seed rain from falling into the trays. One motion sensitive camera was placed at each patch to record animal activity around a single block. The MIT and control were both captured within the frame of the camera.

The distance between the blocks placed on the north and south sides of the patch and between the blocks on east and west sides was measured. A sampling line was established between the two blocks that were the furthest apart. Ten equally spaced sampling points were established along the line. At each sampling point, a 0.25 m2 quadrant was placed a random distance between 0 and 100 cm from the transect line and the number of second-year A. petiolata plants were counted. This was done to estimate the average density of second-year plants per 1 m2.

Trays were placed in the field during peak seed dispersal. In 2013, the trays were in the field from July 3rd to August 7th. In 2014, at South Breens Woods trays were in field from July 2nd to August 8th while at Merwin Nature Preserve trays were out from July 8th to August 8th. After the trays were collected, they were transported to Illinois State University to overwinter outdoors since cold-moist stratification is necessary for seed germination (Baskin & Baskin, 1992). The trays were moved to a heated greenhouse on Feb 20th in 2014 and Feb 16th in 2015. Alliaria petiolata seedlings were counted daily until no new seedlings were observed on two consecutive days, because by this time 95% of the trays had no new seedlings for five consecutive days. Counting was terminated on March 22nd and 12th in 2014 and 2015, respectively. The majority of seeds within the trays are likely to have germinated since 70% of A. petiolata seeds germinate the first year. Also, there is no known reason why germination rates would differ between the MIT and control trays.

The number of animal visits in the photos recorded by the motion sensitive cameras was counted for each treatment. An animal was considered to have visited the MIT treatment if it entered the frame, while a visit to the control was counted if an animal touched the outside of the frame. Photos were analyzed using a chi-square analysis to determine if there was a significant difference in animal visits between the treatments. The A. petiolata seedling counts in the germination trays were analyzed with a mixed linear model (PROC MIXED) to test for a significant difference between the control and MIT. Treatment was a fixed effect while block, block nested within A. petiolata patch, and year were included as random effects in the model. The data were square root transformed to meet the assumptions of normality. All statistical tests were performed in SAS® software 9.3 (SAS Institute, 2012). Alpha levels were set at 0.05 for all tests. This project was approved by the Illinois State University Institutional Animal Care and Use Committee (IACUC). The IACUC number is 14-2013.

Seed attachment

The attachment potential of A. petiolata seeds was measured using a white-tailed deer and a raccoon pelt. Both of these animals are common within the study sites and across North America. The pelts consisted of the skin of the animal with the fur still attached. The deer fur consisted of 2–3 cm long hairs that were flattened from the front of the animal towards the back. The raccoon fur had 5–6 cm long hairs with many smaller hairs, less than 4 cm underneath forming a thick undercoat. Both hair types generally stood upright. The pelts were placed between two wood boards with a 25 × 25 cm opening leaving that area of fur uncovered. The two boards were clamped together to secure the pelts. A 9 × 16 cm grid of 144, 2 × 2 cm squares was centered 15 cm above the fur in a horizontal position with the fur side up. In each trial, 100 A. petiolata seeds were dropped singly through randomly selected squares onto the fur. The pelt and frame were then rotated 180 degrees over a collection box and then immediately turned back to the original position. The number of seeds that remained attached to the fur were counted. Attachment potential was measured as the proportion of seeds that remained attached to the furs after they were rotated. To determine if attachment potential differed between dry and wet fur, furs were misted with 40 ml of water using a plastic spray bottle before the seeds were dropped. The moisture of seeds was also manipulated by partially submerging the seeds in water before they were dropped onto the fur.

For each fur type (racoon or deer), ten replicates trials were used for each of the four treatment combinations, seed dry and fur dry (SD/FD), seed dry and fur wet (SD/FW), seed wet and fur dry (SW/FD), and seed wet and fur wet (SW/FW) for a total of 40 trials per fur type. The raccoon and deer furs were analyzed separately.

To test for a significant effect of seed condition, fur condition and their interaction, the data were aligned and rank transformed (ART) since they could not be transformed to meet assumptions of a parametric Analysis of Variance (ANOVA). Data were aligned by removing the marginal means of all other factors from the response variable other than the factor for which the alignment was being applied (Wobbrock et al., 2011). For example, to analyze the interaction effect of a two-way factorial, the marginal means of the main effects are removed from each response variable to isolate the interaction effect. The aligned data were then ranked, and a two-way ANOVA (PROC GLM) was performed on the ranks. Separate ANOVAs were performed for each main effect and the interaction. For a significant interaction, a Tukey post-hoc analysis was performed. The data were aligned and ranked using the ARTool (Wobbrock et al., 2011). The ART is an appropriate alternative to parametric F-tests when analyzing factorial designs (Mansouri, Paige & Surles, 2004). The ART is robust to Type 1 error (Mansouri, 1999) and has greater power than parametric F-tests when normality assumptions are not met (Richter, 1999).

Seed retention

The same deer and raccoon pelts were attached to separate 25 × 38 cm sections of cardboard. Before seeds were attached, the furs were homogenized by combing the furs two times horizontally and vertically using a plastic comb with 4 cm long teeth spaced 0.9 cm apart. A 5 × 10 grid of 2 × 2 cm cells was placed over the furs and two seeds were dropped per cell from a height of 2 cm. Seeds were then combed into the fur with the same method as homogenization. This procedure is similar to previous epizoochory studies (Römermann, Tackenberg & Poschlod, 2005; Tackenberg et al., 2006; De Pablos & Peco, 2007). The furs were rotated 180 degrees over a collection box to collect the seeds that did not attach. Next the furs were clamped to a collection bin that was attached to a Fisher Vortex Genie 2, which shook the fur and bin horizontally for 1 h. The Fisher Vortex abruptly moved the furs back and forth 0.5 cm. The numbers of horizontal movements were counted for 1 min during the first minute, 30th minute, and 59th minute to ensure that each trial had between 145 to 155 movements per minute. To test for the effect of moisture, furs were misted with water with the same process as described in the attachment potential experiment after the seeds were combed into the fur. There were five trials for each fur by moisture combination.

Other studies (Römermann, Tackenberg & Poschlod, 2005; Tackenberg et al., 2006; De Pablos & Peco, 2007) used a specialized shaking machine that was able to shake furs horizontally and vertically to test for an effect of position on seed retention potential. We were unable to test the effect of fur position since the Fisher Vortex Genie 2 is only capable of moving furs horizontally. However, the results of this study are likely comparable to other studies since fur position was found to have no effect on retention potential (Tackenberg et al., 2006), or only an effect for cattle fur (De Pablos & Peco, 2007), which was not used in this study.

Retention potential was measured as the proportion of seeds that remained attached after 1 h of shaking. For each pelt type, a two-sample t-test (PROC TTEST) was done to determine if the retention potential was significantly different between wet and dry fur. Unequal variances were assumed and the Satterwaite’s test was used as an alternative to the Student’s t-test (Ruxton, 2006). The mean retention potential was considered significantly greater than zero if the 95% confidence intervals did not overlap with zero.

Results

Dispersal kernels

The estimated number of seeds released from the three point sources was 4,012, 4,020 and 4,815 for Point Sources 1, 2, and 3, respectively. The total number and percentage of seeds captured from the point sources was 384 (9.57%), 629 (15.65%), and 682 (14.16%) for Point Sources 1, 2, and 3, respectively. For all three point sources, the mean number of seeds captured per trap was highest in traps placed at distance 0.25 m, and this number decreased as distance from the point source increased (Table 2). Point Source 1 had the lowest mean number of seeds per trap at distance 0.25 m with 23.4 ± 6.47 seeds per trap and Point Source 2 had the highest with 49.6 (±10.51). A small number of seeds was dispersed 2.25 m with all three point sources averaging less than one seed per trap. Even fewer seeds were dispersed 3.25 m with all point sources averaging below 0.5 seeds per trap (Table 2).

Table 2 The average number of seeds (±SE) captured in a single trap at each distance for all three point sources.

D (m)	Source 1	Source 2	Source 3	
0.25	23.4 (±6.47)	49.6 (±10.51)	39.7 (±11.32)	
0.5	11.2 (±5.52)	16.7 (±3.27)	26.0 (±6.64)	
0.75	4.8 (±1.44)	7.8 (±1.24)	7.9 (1.96)	
1.25	1.6 (±0.43)	1.3 (0.36)	0.46 (±0.210)	
2.25	0.84 (±0.147)	0.22 (0.088)	0.16 (±0.066)	
3.25	0.46 (±0.104)	0.19 (0.063)	0.12 (±0.054)	

The AIC scores of the eight g(r) dispersal functions ranged from 1,008.8 to 1,033.5. The Weibull function had the lowest AIC score, but the g(r) and f(r) functions did not integrate to one. Because the Weibull did not meet the pdf requirement, the lognormal function was selected next for analysis since it had the next lowest AIC score at 1,020.4 and the g(r) and f(r) met the requirements of a pdf. However, the g(r) of the lognormal function predicted that the probability density of a seed dispersing into an infinitely small area was zero at distance zero. This prediction was in direct contradiction with field observations that many seeds fall directly below the parent plant, which should result in the density probability being greater than zero at distance zero. Because of this unrealistic prediction, the lognormal function was not used for further analysis.

The 2Dt kernel had the next lowest AIC score after the lognormal at 1,025.5 and the g(r) and f(r) met the requirements of a pdf. The 2Dt g(r) kernel predicted that the probability density of a seed landing in an infinitely small area is highest at distance zero and then steadily declines until 1 m (Fig. 2). This result is more in agreement with field observations and is different than the lognormal. Beyond 0.25 m, the g(r)’s probability densities of the 2Dt rapidly declined as distance increased to 1 m. As the distance increased beyond 1 m, the probability density asymptotically approached zero.

Figure 2 The density pdf (g(r)) of the 2Dt function.

The g(r) describes the probability of a seed landing into an infinitely small area at a specific distance from the point source.

The 2Dt g(r) function was placed into (2) to predict the seed counts per trap for each of the point sources separately. The 2Dt function predicts that the seed count per trap is highest at 0 m and then the predicted count steadily decreases until 1.30 m, where less than one seed per trap is predicted. The predicted seed count per trap continues to decrease beyond 1.30 m asymptotically approaching zero (Fig. 3).

Figure 3 The predicted seed count per trap (solid line) ± 95% confidence intervals (dashed lines) for 2Dt function for each point source, (A) Point Source 1, (B) Point Source 2, and (C) Point Source 3.

Each of the point sources was plotted separately and the black diamonds are the seed counts from the seed traps.

The predicted change in dispersed seed density from a single second-year plant as predicted by the 2Dt g(r) function differs from the prediction of the negative exponential function from Eschtruth & Battles (2009). Specifically, the negative exponential function predicts dispersed seed density to be higher than the 2Dt function at 0.5 m and beyond from the point source (Fig. 4).

Figure 4 The change in predicted seed density as distance from the parent plant increases as predicted by the negative exponential function from Eschtruth & Battles (2009) and the 2Dt g(r) function.

The fecundity of the parent plant was set to 156 seeds as this was the value used by Eschtruth & Battles (2009). Beginning at 0.50 meters, the negative exponential overestimates the dispersed seed density compared to the 2Dt g(r) kernel.

The corresponding f(r) of the 2Dt function has a probability density of zero at distance zero, which is a condition any f(r) will meet due to the multiplier r equaling zero at distance zero in (1). The probability density of the 2Dt function rapidly increases between 0 and 0.25 m and peaks at 0.35 m, meaning seeds have the highest probability of dispersing this distance (Fig. 5). The probability density then steadily declines to around 1.20 m and asymptotically approaches zero beyond that distance. The median, mean, and the distance at which 95% of seeds were dispersed within are 0.47, 0.53 and 1.14 m.

Figure 5 The distance pdf (f(r)) of the 2Dt function.

The f(r) describes the probability of a seed dispersing to a specific distance from the point source.

The parameter values for the 2Dt function were based on the pooled data of the three replicate plots (Table 3). SAS approximates the standard errors, P-values are for alpha <0.05 and hypothesis of parameter = 0. Variation in seed counts within traps was not significantly different among the three point sources as the parameter u was not significantly different than zero. The parameter k was less than one, which indicates that there was a high amount of variation around the expected seed trap values (Clark et al., 2005). The high variation is apparent when comparing the predicted seed counts per trap of the dispersal functions to the actual seed counts from the traps in the field (Fig. 3). There was a large amount of variability in the number of seeds captured at distances 0.25 and 0.5 m. At the 0.25 m distance, captured seeds varied from three seeds to 117 seeds per trap, and from two to 59 at 0.5 m distance.

Table 3 The parameter estimates and their standard errors of the 2Dt function.

Parameter values were based on the pooled data from the three plots. Standard error values are approximate due to the method SAS uses to calculate them. P-values are for alpha <0.05 and hypothesis of parameter = 0.

Parameter	Estimate	St. Err	DF	p-value	
A	1.0561	0.3427	2	0.0911	
B	4.8795	2.1794	2	0.1545	
U	0.1474	0.1282	2	0.3692	
K	0.6493	0.1062	2	0.0257	

Epizoochory field experiment

The bait placed in the MIT was removed daily for most pie pans in both years, indicating animals were visiting the treatments with a high frequency. This high level of animal activity at the MIT was supported by the photos from the motion sensitive cameras. For both years and all A. petiolata patches combined, the MIT had 951 animal visits, which was significantly greater (χ1,9822=788.6, p < 0.0001) than the 51 visits to the controls. Raccoons accounted for most animal visits and were the only animal recorded at all seven A. petiolata patches (Table 4). Raccoons entered the MIT wood frames and stood directly over the germination trays while feeding. Turkeys (Meleagris gallopavo) were the second most common animal visitor, but they were only recorded in the year 2014 and only at the South Breen Woods. Turkeys and deer were photographed eating the bait by inserting only their head into the open end of the frame. The increased animal activity over the MIT germination trays resulted in significantly more (F1,27 = 129.5, p < 0.0001) A. petiolata seedlings than in control trays with average MIT counts more than one order of magnitude greater than for the control trays (Fig. 6).

Table 4 The photo counts for each animal species that visited the treatments for both study years combined.

Turkeys were only observed in 2014. The raccoon was the only animal observed at all seven A. petiolata patches.

Animal	MIT	Control	
Raccoon	720	46	
Turkey	147	0	
Deer	32	0	
Squirrel	2	2	
Nuthatch	9	0	
Blue Jay	3	0	
Chipmunk	10	2	
Mouse	4	0	
Cardinal	1	0	
Mourning Dove	3	0	
Woodchuck	1	0	

Figure 6 The back transformed mean number (±95% CI) of A. petiolata seedlings counted in the germination trays of the two treatments.

The confidence intervals are not symmetrical because of the back transformation. The MIT trays (p < 0.0001) had significantly more seedlings than the control trays.

The seven A. petiolata patches used in this experiment varied in patch size and in density of second-year plants (Table 5). While the A. petiolata patches differed in size, there appeared to be no pattern to the variation unlike second-year plant density. All patches from summer 2014 had lower second-year plant density than patches from 2013. However, this difference in density did not affect A. petiolata seedling counts in the germination trays. The random variation attributable to A. petiolata patch and year to seedling counts were not significant (p > 0.10 for both). The variation due to block was estimated to be zero; therefore, SAS PROC MIXED did not test for significance.

Table 5 The distance (m) between the north and south blocks and the east and west blocks for each A. petiolata patch and the average density of second-year plants per 1 m2.

Patches 1–3 and 4–7 were used in 2013 and 2014, respectively.

Patch	N to S (m)	E to W (m)	Density (m2)	
1	10	16	141	
2	9.4	9.4	237	
3	22	11.8	258	
4	11.3	14.5	41	
5	13	11.3	51	
6	18.3	13.3	24	
7	20	17	60	

Seed attachment

For the deer pelt treatment, the main effects of fur (F1,39 = 56.44, P < 0.0001) and seed (F1,39 = 110.3, P < 0.0001) conditions, and their interaction (F1,39 = 59.8, P < 0.0001) significantly affected attachment potential. Fur (F1,39 = 3920.4, P < 0.0001) and seed conditions (F1,39 = 100.39, P < 0.0001), and their interaction (F1,39 = 81.29, P < 0.0001) also significantly affected attachment potential on the raccoon pelt. Seed attachment potential was highest for both pelt types when seeds were wet, regardless of fur condition (Fig. 7A). When seeds were dry, more seeds attached to wet fur than dry fur.

Figure 7 The mean (±SD) AtP values for each treatment combination for the (A) deer and (C) raccoon pelts and the box and whisker plot of results of the Tukey follow test on the ART values of the interaction between seed and fur condition for the (B) deer and (D) racoon.

Significant differences are marked by different letters.

The Tukey follow-up test of the interaction term found significant differences in the ART ranks. For both pelt types, the SD/FW and the SW/FD treatments had significantly higher ranks than the SD/FD and the SW/FW treatments (Fig. 7B). For the SW/FD treatment, the weak effect on attachment potential of the dry fur was overcome by the wet seed resulting in a high attachment potential. For the SD/FW treatment, the weak effect of the dry seed was overcome by the effect of the wet fur resulting again in a high attachment potential. Low ART values for SW/FW treatment, despite high attachment potential values, were due to strong positive effects on attachment potential of both wet fur and wet seeds, with no increase in attachment potential in the combined treatment. Thus, high attachment potential values will result, if either the seed or fur is wet.

Seed retention

The retention potential was significantly greater for wet deer fur (T4.63 = 29.6  p < 0.0001) and wet raccoon fur (T7.27 = 74.78, p < 0.0001) when compared to dry fur (Fig. 8). The difference in retention potential between wet and dry fur was large for both pelt types. For raccoon fur, values ranged from 2–5% for dry fur compared to 94–100% for wet fur; for deer fur, values ranged from 1–6% for dry fur from 81–98% for wet fur. While the retention potential was significantly lower for dry fur, it was still significantly greater than zero since the 95% confidence intervals did not overlap with zero.

Figure 8 The mean (±95% CI) retention potential for the deer and raccoon pelts.

The retention potential was significantly higher when the pelt was wet for both the deer and raccoon pelts. The 95% confidence intervals did not overlap with zero for any treatment.

Discussion

The results from this experimental study provide a more accurate and precise prediction of dispersal distances in A. petiolata than those available in the literature (Nuzzo, 1999; Drayton & Primack, 1999), although the estimates from prior studies predict that most seeds fell within 1–2 m of the parent plants are consistent with our results. The mean and median seed dispersal distances predicted by the 2Dt function were about 0.50 m, which is substantially less than the 1.28 m (range 1.03–1.63 m) predicted by Biswas & Wagner (2015) who used an experimental design similar to ours. However, the first seed traps in that study were placed 0.50 m from the point seed source which may have resulted in the majority of dispersed seeds being missed as our results indicate peak seed dispersal occurred at 0.35. This likely caused the average dispersal to be overestimated.

The 2Dt function predicted that the distance at which 95% of A. petiolata seeds are dispersed is about 1.14 m, which is substantially less than the 2 m used to estimate the value of the b parameter of the negative exponential function used by Eschtruth & Battles (2009), Eschtruth & Battles (2011) and Eschtruth & Battles (2014) suggesting an overestimation of seed dispersal distances in the studies. The overestimation of dispersal distances is also apparent when the predicted dispersed seed density of the negative exponential function is compared to that of the 2Dt function (Fig. 4). The negative exponential overestimates dispersed seed density at distances greater than 0.50 m. By overestimating dispersal distances, the seed rain index of Eschtruth & Battles (2009), Eschtruth & Battles (2011) and Eschtruth & Battles (2014) also overestimated the seed rain entering their research plots, which resulted in an over estimation of propagule pressure. Incorporating the experimentally based dispersal functions from this study will improve the accuracy of estimates of seed rain, and therefore, propagule pressure.

With only 5% of A. petiolata seeds being dispersed over 1.14 m (Fig. 3), A. petiolata is similar to most plant species in that the vast majority of seeds are dispersed within a short distance from the parent plant (Wilson, 1993; Kot, Van Lewis & van, 1996; Venable et al., 2008) with only a small proportion dispersed long distances (Cain, Milligan & Strand, 2000; Nathan, 2006; Nathan et al., 2008). However, these relatively rare long-distance dispersal events are more important to the spread of a species across the landscape that many short distance dispersal events (Clark et al., 1998; Suarez, Holway & Case, 2001; Nathan, Cain & Levin, 2003; Theoharides & Dukes, 2007; Pergl et al., 2011). Our results indicate the epizoochory is likely one mechanism by which seeds can be dispersed greater distances.

This study is the first to provide experimental evidence that epizoochory through woodland animals is a potential seed dispersal mechanism of A. petiolata. The MIT germination trays had significantly more animal visits than the control trays (Table 4) which resulted in the MIT trays having significantly more A. petiolata seedlings (Fig. 6). The laboratory studies provide evidence that seeds can adhere to raccoon and deer fur sufficiently for dispersal (Figs. 7 and 8). However, these results do not rule out the possibility of seeds being dispersed by attachment to hooves, paws, or claws (Gill & Beardall, 2001; Heinken et al., 2006; Schulze, Buchwald & Heinken, 2014). Since attachment and retention potential increased when seeds or fur were wet (Figs. 7 and 8) it is likely wet environmental conditions, such as rainfall or heavy dew, increase A. petiolata epizoochory potential.

Alliaria petiolata seeds that are retained within deer and raccoon fur have the potential to be dispersed several kilometers by these animals. The longer a seed is retained in the fur of an animal, the farther it can be dispersed by that animal (Couvreur et al., 2005; Adriaens, Honnay & Hermy, 2007; Guttal et al., 2011), particularly with larger home ranges. The home range size of deer can range from less 1 km2 to more than 10 km2 depending on season and age of the deer (Lesage et al., 2000). The home range size of raccoons can range from less than 0.5 km2 to more than 1 km2 depending on resource availability and season (Gehrt & Fritzell, 1998; Beasley, Devault & Rhodes, 2007).

While A. petiolata seeds lack clear adaptations for epizoochory, other studies have also found that seeds without special adaptations for animal dispersal exhibit epizoochory, albeit at a lower proportion of total seed production compared to plant species with specific adaptations (Fischer et al., 1996; Couvreur et al., 2004; Hovstad, Borvik & Ohlson, 2009). A lack of adaptations by A. petiolata may be compensated for by high seed production (Anderson, Dhillion & Kelley, 1996; Nuzzo, 1999; Susko & Lovett-Doust, 2000; Will & Tackenberg, 2008; Couvreur et al., 2008). Additionally, autogamy in A. petiolata can allow establishment of new populations from a small number of dispersed seeds (Anderson, Dhillion & Kelley, 1996). Therefore, epizoochory may be an important mechanism for the spread of A. petiolata across the landscape accounting for rates of expansion greater than predicted rate of less than 1 m per year (Nuzzo, 1999). Epizoochory may also contribute to A. petiolata’s invasion success as it may increase the probability that seeds are deposited on favorable microsites within woodlands (Nathan & Muller-Landau, 2000).

Endozoochory (seed dispersal in animal guts) is another common seed dispersal mechanism, but it is highly unlikely that that it is a dispersal mechanism of A. petiolata. Alliaria petiolata experiences very little herbivory (Evans & Landis, 2007; Van Riper, Becker & Skinner, 2010) due to production of toxins in plant tissues such as cyanide and glucosinolates (Barto, Powell & Cipollini, 2010; Cipollini & Gruner, 2007). Hydrochory (seed dispersal through water) has been suggested as a seed dispersal mechanism of A. petiolata due to it being prevalent in floodplain areas (Nuzzo, 1999; Meekins & Mccarthy, 2001). While hydrochory may occur, this does not explain how A. petiolata spreads into upland areas (Burls & McClaugherty, 2008), spreads up stream (Nuzzo, 1993), or disperses locally across the landscape (Eschtruth & Battles, 2009). Hydrochory also does not explain the A. petiolata seedling differences between the MIT and control germination trays, because the two treatments did not experience any differences in the flow of water from the ground surface into the trays.

Causes for increased attachment retention potential, and attachment potential, when the seed or fur are wet is unclear. Some plant species produce seed coat mucilage when wet increasing epizoochory (Yang et al., 2012). When A. petiolata seeds were observed under a light microscope, they did not appear to produce any mucilage when wet. Another possibility is that the water forms hydrogen bonds between the seeds and fur which increases the retention and attachment potential. However, this idea was not explored in this study and further research is needed to understand the role of water as a dispersal agent for A. petiolata.

Another topic that needs further clarification is the role genetic and environmental variation plays in A. petiolata seed dispersal. Byers & Quinn (1998) found that certain A. petiolata traits, such as seed mass, differed among studied habitats. Susko & Lovett-Doust (2000) reported that A. petiolata seed mass was highly variable among and within populations. It is unknown how the variability in such traits as seed mass may affect seed dispersal distances. Environmental effects on A. petiolata also need to be studied as other studies have found that such factors as habitat type affects dispersal distances (Fontúrbel, Jordano & Medel, 2017). It will be important to study these factors to further improve seed dispersal estimates of A. petiolata.

We thank the ParkLands Foundation for allowing us to conduct this research on their property and Victoria A. Borowicz and Scott K. Sakaluk for assistance with this research.

Additional Information and Declarations

Competing Interests

Author Contributions

Animal Ethics

Data Availability

The authors declare there are no competing interests.

Christopher A. Loebach conceived and designed the experiments, performed the experiments, analyzed the data, contributed reagents/materials/analysis tools, prepared figures and/or tables, authored or reviewed drafts of the paper, approved the final draft.

Roger C. Anderson conceived and designed the experiments, performed the experiments, analyzed the data, contributed reagents/materials/analysis tools, authored or reviewed drafts of the paper, approved the final draft.

The following information was supplied relating to ethical approvals (i.e., approving body and any reference numbers):

This project was approved by the Illinois State University Institutional Animal Care and Use Committee (IACUC). The IACUC number is 14-2013.

The following information was supplied regarding data availability:

Figshare: https://figshare.com/projects/Measuring_short_distance_dispersal_of_Alliaria_petiolata_and_determining_potential_long_distance_dispersal_mechanisms_/19189

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
