# Peer review of "Measuring short distance dispersal of Alliaria petiolata and determining potential long distance dispersal mechanisms"

_PeerJ, doi:10.7717/peerj.4477_

## Round 0.1 · original submission · Minor Revisions

I have comments from three reviewers and, as you'll see, they are divergent in some aspects of their assessment. All find the research question important, and basic premise and approach interesting and appropriate for PeerJ. Reviews differ, however, in their judgment of the presentation of design and methods, and the robustness of the data-set with regard to conclusions reached.

I am asking you to undertake a revision to address the specific issues and suggestions raised by reviewers. Most importantly, work to a) clarify methods -- both field/experimental and statistical -- where reviewers note ambiguity, and b) acknowledge and assess limitations of the data-set with respect to questions raised, particularly, by reviewer three. That reviewer notes some of the sorts of qualifications and reservations that might be important to acknowledge (indeed, exploring some of these as gnerating research questions may add a nice dimension to the paper). I think careful address of these issues can bring the manuscript in line with PeerJ's acceptance criteria.

Thanks for your submission, and I look forward to the revision.

Reviewer 1 ·

Basic reporting

- Well-written manuscript, clearly presented with sufficient literature references and relevant, interesting results presented
- Please provide REF for lines 64-66
- Line 82, unclear which untested assumption is being referred to here
- Line 87, “untested observations” is a strange term – observations or untested assumptions?
- It would be helpful to condense and re-order the information in lines 77-93 to avoid repeating statements and creating confusion
- Lines 100 and 101 are slightly repetitive – could be combined or re-worded to simplify
- Need REF and location for specific dates on line 164 – this is different in different places
- Need location for seed mass and seed dispersal weights and dates provided

Experimental design

- Research objectives are original and within the aims and scopes of PeerJ
- Please provide climate data (MAT & MAP), soil types, as well as invasion history of sites if available
- How shaded were the areas used for the seed trapping? Were they a certain distance from the forest edge? Unclear how they were they chosen besides having “minimal understory vegetation and nearly level topography”.
- What was the understory vegetation? Other invasives present?
- Including a diagram of the trapping experimental design (lines 194-217) would be really helpful – more helpful than Figure 1, which doesn’t seem to be necessary in my opinion
- Were any tests done to confirm that seed dispersal was isotropic, as was assumed?

Validity of the findings

- Although I am not versed in the specific kernel analyses that the authors did for this study, the findings appear statistically sound, controlled, and valid.
- Conclusions are clearly stated.
- Might be helpful to make a mention of species-specificity in the paragraph for lines 525-537.
- How much variation in seed rain and dispersal distance happens within species – is this effect population-dependent or can it be applied to all A. petioltata?
- Is there any work on A. petiolata for seed dispersal in its native range? Can you make any larger conclusions about why this animal fur seed dispersal may be (at least in part) responsible for its invasive success?
- It would be helpful to work into both the Intro & the Discussion some larger picture scenarios. What does this mean for other invasive plants? How might this impact the spread of this species large-scale?

Additional comments

This is a well-written and ecologically interesting manuscript. It is a well-designed experiment and provides valuable knowledge to invasion ecology. My suggestions for revisions are within the above boxes.

·

Basic reporting

All good.

Experimental design

Sound.

Validity of the findings

Data analysis was sound.

Reviewer 3 ·

Basic reporting

Line 38-42, insufficient explanation of the experimental design (Mammal inclusion)
Lines 43-46, ditto for mammal fur test

Figures showing mammal inclusion treatments are not very informative and can be removed

Line 65-67 makes a broad statement about competitive & invasive ability with no citations

First paragraph not connected to main objectives of the study and could be condensed to 1-2 sentences or omitted. Better would be a more general paragraph about seed dispersal life history strategies in plants in general, and how it might affect predictions about invasion.

More compelling would be to introduce the idea that dispersal distances used in modeling to date have been based on cursory field observations.

Lines 150-172 are a nice introduction to the study species, much better than the first paragraph

Lines 190-192 repeat information from above about the site

Methods on field measurement for dispersal kernel data are very vague. Line 196 what is a point source and how many were there? Were there 15 plants in total or multiple sets of 15, and if the latter, how was spatial variation considered?

Lines 233-281 description of the dispersal kernel modeling could be considerably condensed. It is thorough and reasonably clear but the text needs to be tightened up. Again, is this N=15 or were there multiple sites with 15 and if so, how many?

Figures 1 and 2 are not very helpful. The description in the text is sufficient.

Line 349-357, more detail is needed to understand the design – I had to read this a few times and I am not sure if I understand: there were 100 seeds per pelt and 10 trials per pelt? Please revise for clarity of methods/design.

ANOVA’s –apostrophe inappropriate here

Experimental design

The experimental design is not clear and appears to be both under-replicated and to have some flaws.

Only 15 plants were used to estimate dispersal distance; intro sets the expectation of experimental data for dispersal distance but design falls a bit short on delivering this type of data.

Only one site was used, how might this affect the findings? i.e., genetic & envirionmental variation could select for different dispersal modes.

Design for attracting mammals to the seed trays is unclear and may influence the overall conclusions due to design flaw effects. As I understand it food was placed as an attractant to mammals, correct? If so, how might this affect overestimates of mammal interactions with garlic mustard, given an increase in mammal visitation rates to invaded plot?

Validity of the findings

The premise of the work is solid - we need better field and experimental data on garlic mustard seed dispersal. However, while I agree that this study has the potential to contribute new data on dispersal distances and mechanisms in the study species, it is difficult to determine from the current version whether or not the design is sufficient to support the findings.

Additional comments

The goals of this paper are to quantify dispersal distances in garlic mustard and to test the potential for animal dispersal of its seeds, which are classically thought to be passively dispersed. The authors use a combination of passive seed traps in the field, experiments to test for seed bank accumulation with and without potential animal traffic, and a simple test of whether garlic mustard seeds can attach to animal pelts (both wet and dry fur).

While such data are lacking for this invasive plant and may be useful for making better predictions about its spread, the manuscript as written is unclear as to whether the design is sufficient for such interpretation. In general it appears that the sample sizes are very small, that some aspects of the design (i.e., phenotypic and environmental variation; attracting more animals than would be normally found in the field in garlic mustard patches) and that each element of this paper requires additional replication and development.

---

## Round 0.2 · Minor Revisions

I have decided that further peer review is not called for -- I judge that you've addressed most of the previous reviewers' comments appropriately. However, after a closer reading of the MS, I'm asking you to undertake another round of minor revision before acceptance.

I've attached a copy of the paper with my suggestions and comments inserted using MS-Word 'track changes'. Most of these concern simple issues -- typos, minor issues with syntax, etc. -- but there are quite a few of these. I've also noted instances of repetition or inefficient prose with some suggested rephrasings. In a few cases, particularly where things were unclear, I've inserted questions or comments in italic font within square brackets; address these as you think best.

Giving attention to these sorts of issues (and there may be similar instances that I've missed) will, I think, make a more compact and readable paper.

---

## Round 0.3 · accepted · Accept

Thanks again for your work in revision.